# External particle mixing influences hygroscopicity in a sub-urban area

Shravan Deshmukh[1], Laurent Poulain[1], Birgit Wehner[1], Silvia Henning[1], Jean-Eudes Petit[3], Pauline Fombelle[3,a], Olivier Favez[4], Hartmut Herrmann[1], Mira Pöhlker[1,2]

[1]Leibniz Institute for Tropospheric Research, e.V. (TROPOS), Permoserstrasse 15, 04318 Leipzig, Germany
[2]Faculty of Physics and Earth Sciences, Leipzig Institute for Meteorology, Leipzig University, 04103 Leipzig, Germany
[3]Laboratoire des Sciences du Climat et de l'Environnement, CEA-CNRS-UVSQ, IPSL, Université Paris-Saclay, 91191 Gif-sur-Yvette, France
[4]Institut National de l'Environnement Industriel et des Risques, Parc Technologique ALATA, Verneuil-en-Halatte, France
[a]now at : Université Clermont Auvergne, Laboratoire de Météorologie Physique, OPGC/CNRS UMR 6016, Clermont-Ferrand, France

*Correspondence to*: Shravan Deshmukh (deshmukh@tropos.de), Laurent Poulain (poulain@tropos.de) and Mira
Pöhlker (poehlker@tropos.de)

**Abstract.** Hygroscopicity strongly influences aerosol properties and multiphase chemistry, which is essential in several atmospheric processes. Although CCN (cloud condensation nuclei) properties are commonly measured, sub-saturated hygroscopicity measurements remain rare. During the ACROSS campaign (**A**tmospheric **C**hemist**R**y **O**f the **S**uburban fore**St**, in Paris, Summer 2022), the particle's hygroscopic growth at 90 % relative
humidity (RH) and chemical composition were measured at the sub-urban site using a Hygroscopicity Tandem Differential Mobility Analyser (HTDMA, scanning at 100, 150, 200, and 250 nm) and an Aerodyne High-Resolution Time-of-Flight Aerosol Mass Spectrometer (HR-ToF-AMS). Growth factor probability density functions (GF-PDFs) revealed two distinct modes—hydrophobic and hygroscopic—suggesting a combination of internal and external particle mixing with the split at GF 1.2. The prevalence of the hygroscopic mode increased
with particle size, with mean hygroscopicity (κ) values of 0.23 and 0.38 for 100 and 200 nm particles, respectively. Using the Zdanovskii-Stokes-Robinson (ZSR) mixing rule, the agreement between measured and chemically derived hygroscopicity was approximately 51% for 100nm particles, which declined for 200 and 250 nm. These emphasise the large effect of external particle mixing and its influence on predicting hygroscopicity. The ZSR approach proves unreliable in predicting the wide growth distribution of externally mixed particles. In this
measurement, 80-90% of the particles were externally mixed and influenced by fresh emission, which affected the hygroscopicity prediction by a factor of 2. A cluster analysis based on backward trajectories and meteorological conditions gives valuable insights into the chemical composition and variations in the hygroscopicity of different air masses.

## 1 Introduction

The atmospheric aerosol significantly impacts the Earth's radiation budget and climate on regional and global scales, exerting direct and indirect effects (IPCC 2013; Rosenfeld et al., 2014; Li et al., 2016). Aerosols affect cloud formation through their hygroscopic properties and how they are incorporated into climate model simulations. Understanding these interactions is important for precise climate predictions. Hygroscopicity is the ability of particles to absorb water from the environment and is one of the essential physiochemical properties of

aerosols. The hygroscopic growth of aerosol in the sub-saturate regime, where the ambient relative humidity is below 100%, is crucial for discerning their interactions with water vapour in the atmosphere, influencing their size, shape, composition, and involvement in atmospheric processes. Aerosols absorb water and grow in size, impacting their optical properties and ability to form cloud droplets, influencing both direct and indirect radiative forcing (IPCC 2007 and 2021). These aerosol particles engage with solar radiation through absorption and

scattering, contributing to positive or negative radiative forcing (Haywood and Boucher, 2000).

The hygroscopic growth of aerosols can be measured or estimated through direct and indirect techniques (Hegg et al., 2007; Achtert et al., 2009). The hygroscopic tandem differential mobility analyser (HTDMA) is used as a real-time direct measurement technique for fine-mode aerosols. It gives the growth factor (GF) of particles at a given dry particle diameter and relative humidity (RH). The hygroscopic parameter $\kappa$, a simplified model

parameter representing the composition's dependence on the solution water activity, can be calculated from the measured GF (Petters and Kreidenweis, 2007). $\kappa$ values for highly hygroscopic aerosols, such as sea salts and sulphates, range between 0.5 and 1.4, while non-hygroscopic aerosols, like soot, have values close to zero (Gysel et al., 2007; Bezantakos et al., 2013). A common approach to conducting long-term measurements of aerosol hygroscopicity is to use CCN counters (Paramonov et al., 2015; Schmale et al., 2018), which is quite popular but

used at supersaturated regimes. HTDMAs are employed for sub-saturate regimes. A limited number of long-term studies using the HTDMA technique have been published (Kammermann et al., 2010; Fors et al., 2011). These available studies give insights into the connection between particle hygroscopic growth, chemical composition, and the processes involved in their formation and transformation. HTDMA measurements can improve knowledge on water uptake and RH effect in non-cloud conditions for aerosols, and this can also be useful to improve the

prediction of light absorption and scattering, which directly affect Earth's energy budget and climate. Hygroscopicity primarily depends on the aerosol size and chemical composition (Gysel et al., 2007; Gunthe et al., 2009).

The output of Aerosol Mass Spectrometry (AMS) measurements gives detailed information on organic molecular fragments (Kanakidou et al., 2005) and the chemical composition of aerosol particles. Organic compounds

constitute a substantial but variable fraction of atmospheric aerosols. In contrast to inorganic species, which exhibit well-characterized hygroscopic growth, our understanding of the water uptake of the organic aerosol fraction is limited, which contributes 30-70% of fine particles (Zhang et al., 2007; Hallquist et al., 2009). Note that AMS instruments can only detect particle material that can be volatilized at temperatures up to 600 °C (DeCarlo et al., 2006). Disregarding the non-detectable refractory fraction, which includes widely abundant

elemental carbon and sea salt, may lead to an under or overprediction of the hygroscopic growth factor by 25-40% in closure studies (Gysel et al., 2007; Wu et al., 2013). The Aethalometer AE33 or MAAP instrument can measure undetected black carbon and can be incorporated in closure. A common practice in predicting hygroscopic growth involves utilizing known hygroscopic growth factors of specific pure chemical species while assuming the Zdanovskii–Stokes–Robinson (ZSR) mixing rule, as proposed by (Stokes and Robinson, 1966) and

(Zdanovskii, 1948). This rule has demonstrated its applicability mainly for the internally mixed particles when compared to directly measured growth factors of various atmospheric aerosols (Cubison et al., 2008; Petters et al., 2009; Wu et al., 2013; Pöhlker et al., 2023). The climate-relevant characteristics of atmospheric aerosol particles are heavily influenced by their hygroscopicity and mixing state (Kaufman et al., 2002; McFiggans et al., 2006).

The hygroscopicity and aerosol mixing state are vital in comprehending aerosol interactions with the environment and various atmospheric processes. Ambient aerosol is commonly considered to be a heterogeneous mixture of particles and gases with diverse chemical compositions and sizes. It refers to an internal mixture of aerosol when the particles of the same size have a similar chemical composition (Spitieri et al., 2023). In contrast, an external mixture occurs when particles of the same size have distinctly different chemical compositions. Urban and suburban environments typically exhibit external mixtures from fresh local emissions and hygroscopic background aerosols, while marine environments tend to have highly hygroscopic aerosols (Massling et al., 2007; Swietlicki et al., 2008; Enroth et al., 2018; Wang et al., 2018).

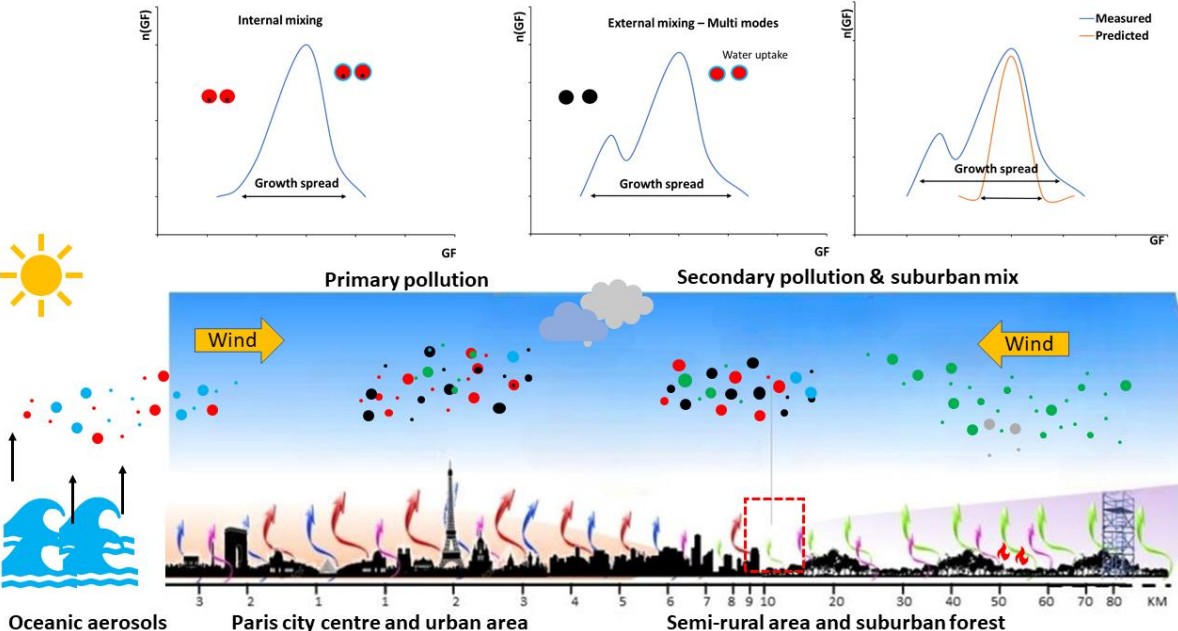

**Figure 1. Illustrative diagram (background picture ACROSS Aeris**, Description – Across, n.d.**) showing the general behaviour of hygroscopicity and particle mixing, with measurement site in a red dotted box during the ACROSS 2022 campaign at LSCE-SIRTA.**

The hygroscopic properties of aerosols change continuously over the lifetime of the particles. This study provides size-resolved hygroscopicity data, which has not been extensively measured often. It aims to understand the ambient aerosol hygroscopic properties with the chemical composition. One month of measurements during the ACROSS campaign provides insights into microphysical parameters, including time and size-resolved HTDMA data under sub-saturated conditions in a suburban environment, shown in Fig. 1. The upward arrows define the processes by which primary pollutants, such as oxides of nitrogen and volatile organic compounds in red, pink and blue colours, respectively, are emitted in the atmosphere, leading to their oxidation and ultimate removal while at the same time producing secondary species, such as ozone and organic aerosols in green colours over suburban to forest region. The hygroscopicity of ambient aerosol was investigated in the particle size range between 100 to 250 nm, providing information about the hygroscopicity and the degree of mixing state for selected particle sizes. The standard deviation or sigma ($\sigma$) of the growth factor derived from the HTDMA can explain the degree of mixing of particles and growth spread (Sjogren et al., 2008; Spitieri et al., 2023); the large $\sigma$ tends to have multiple modes in GF distribution and wide growth spread, which is associated with external mixing and vice versa for internally mixed particles shown on the top figure in the illustrative diagram. Atmospheric measurements were meticulously recorded during the field experiment at the site. Based on these data, our study

provides a characterisation of particle physicochemical properties associated with mixing state and hygroscopic growth. The κ will inform the relationship between the measured and the chemical-derived hygroscopicity. It can improve our understanding of aerosol growth in sub-saturated regimes and their role in the atmosphere,
influencing their direct impact on the aerosol microstructure and optical properties.

## 2 Methodology

### 2.1 Sampling site

Measurements used in this study were performed at the SIRTA observatory (Site Instrumental de Recherche par Télédétection Atmosphérique, http://sirta.ipsl.fr) located approximately 23 km southwest of the Paris city center
(Haeffelin et al., 2005) on the Saclay plateau (2.148° E, 48.708° N; 150 m a.s.l.). This "supersite" is surrounded by suburban facilities, forests, agricultural fields, and roads connecting Paris and provides long-term, in-situ observations of the atmospheric aerosol's chemical, optical, and physical properties since 2011 (Petit et al., 2015). It is part of the European Research Infrastructure for the observation of Aerosol, Clouds, and Trace gases, known as ACTRIS (Laj et al., 2024). Atmospheric composition measurements performed at SIRTA are considered to be
representative of background conditions for the Paris region.

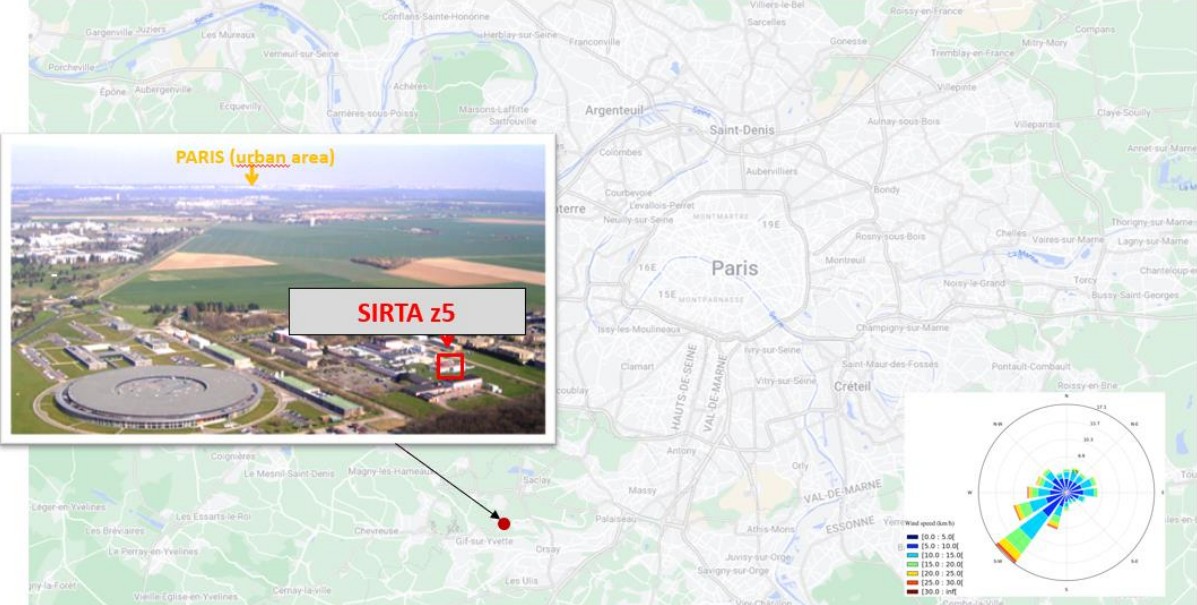

**Figure 2. The SIRTA – LSCE (z5) aerosol measurement station in Paris (from © Google Maps).**

Regarding seasonal features, the summer periods frequently experience pollution episodes mainly related to forest fire and mobile sources (road traffic), and agricultural emissions at a regional scale, together with the transport of
polluted air masses, are associated with mesoscale high-pressure systems (Petit et al., 2014). Standard meteorological parameters were recorded at the station, including temperature, humidity, wind speed, and wind direction. The meteorological sensors were positioned on a meteorological mast at a height of 10 m above ground.

### 2.2 Hygroscopic growth measurements and data inversion

A custom-built humidified tandem differential mobility analyser (HTDMA) by TROPOS was used to measure
the hygroscopic growth factor distributions of ambient aerosol particles (Massling et al., 2007; Wu et al., 2013).

The polydisperse aerosol was first dried by passing through a Nafion aerosol dryer and brought to charge equilibrium by an [85]Kr bipolar neutraliser, shown in Figure S1. The selected dry diameters $D_0$ for this campaign at specific narrow size fractions centred around 100, 150, 200 and 250 nm in a differential mobility analyser (DMA-1) and exposed at relative humidity (RH) of 90 ± 3 % (Bezantakos et al., 2013; Wu et al., 2013). Both

DMAs were operated with a sheath flow rate of 5 l min$^{-1}$ and a sample flow rate of ~1 l min$^{-1}$. It is to be noted that the variance of RH may have a measurable impact on GF, particularly with low-soluble compounds and their sensitivity to RH in this range. This could contribute to GF values uncertainty, especially organics or less-hygroscopic dominant species. Although the RH calibration was maintained carefully, this uncertainty is acknowledged as a potential source of error in our measurements.

The hygroscopic growth factor (GF) determined by the HTDMA is the ratio of the particle mobility diameter, $D_p$ (RH), at a given RH to its dry diameter, $D_0$, in Eq. (1):

$$GF\ (RH,\ D_0) = \frac{Dp\ (RH)}{D_0},\ \ \ \ \ \ \ \ \ \ \ \ \ \ \ \ \ \ (1)$$

The TDMAinv method developed by (Gysel et al., 2009) was used to invert the data. During the entire field measurements, the GF of 100 nm pure ammonium sulphate particles was measured once per four hours at 90 % RH to ensure high-quality data and thus to validate the accuracy and performance of the HTDMA in terms of RH (Gysel et al., 2002). Correspondingly, the relative uncertainty of around ± 3% for GFs of ammonium sulphate measurement is shown in Figure S2.  The residence time of particles at RH (90 %) before entering the DMA-2 is

estimated to be approximately 2.5 seconds in the TROPOS-HTDMA system (Wu et al., 2013). The short residence time could bring an additional bias in the measurements for the particles dominated by organics (Peng & Chan, 2001; Chan & Chan, 2005; Sjogren et al., 2007; Duplissy et al., 2009).

The GF standard deviation or Sigma (σ) of a GF-PDF was determined according to Eq. (C.6) in (Gysel et al., 2009). To assess the variability in growth factors and characterise the mixing state, the sigma is employed as a

metric for the spread of GF ($\sigma_{GF}$), as outlined by (Sjogren et al., 2008). In the present study, the inverted data are categorised into two cases, representative of aerosol mixing states, as illustrated in Fig. S7. Specifically, $\sigma_{GF} \leq$ 0.08 signifies an internally mixed aerosol (Fig. S7a), $\sigma_{GF} \geq 0.10$ characterises an externally mixed aerosol with two distinct modes (Fig. S7b). This categorisation is used to identify the modes in GF-PDF for all scans for all diameters over time. The $\sigma_{GF}$ not only describes the spread of single mode but also gives details of broader

distribution for bimodal. A brief explanation is provided in the supplementary.

**2.3 Particle number size distributions (PNSD)**

The particle number size distributions of ambient aerosol in its dry state were measured using a TROPOS-style MPSS, operated concurrently with the HTDMA and AMS at the SIRTA-LSCE station. It operated with a 5-minute time resolution and an aerosol-to-sheath flow ratio of 1/5 l min$^{-1}$, covering a particle size range from 10 to 850

nm. Both aerosol and sheath flows were dehydrated to a relative humidity lower than 40% using a Nafion dryer. The uncertainty of this instrument's total particle number concentration was determined to be ±15%.

### 2.4 Chemical composition and black carbon measurement

The Aerodyne High-Resolution Time-of-Flight Aerosol Mass Spectrometer (HR-ToF-AMS, referred to as AMS) was typically operated with a time resolution of 2 minutes. The AMS, owing to the vaporiser's 600°C surface temperature (DeCarlo et al., 2006), exclusively analyses the non-refractory chemical composition of particles, making it unable to detect soot, crustal material, and sea salt. Consequently, based on the transmission efficiency of the aerodynamic lens and the identified compounds, the AMS furnishes the chemical composition of the sub-micrometre non-refractory aerosol fraction at NR-PM1 (Canagaratna et al., 2007). Utilising the approach developed by (Aiken et al., 2008) and improved by (Canagaratna et al., 2015), high-resolution organic particle mass spectra were employed to determine the elemental composition and the oxygen-to-carbon (O:C) atomic ratio. The uncertainty of chemical composition for AMS varies for different chemical species, and it is reduced using calibration collection efficiency (CE) for the system. For these measurements, the ion CE was $4.44e^{-08}$, and considering this, the uncertainty for the chemical composition was roughly estimated to be 10-15%.

The concentration of equivalent black carbon (eBC) was measured using a multi-wavelength aethalometer (AE33 model, Magee Scientific, 1-minute time resolution, wavelength $\lambda = 880$ nm) from (Weingartner et al., 2003). For ambient measurements, uncertainties in BC concentrations using the AE33 can range from 5% to 15% (Drinovec et al., 2015). The uncertainty during this measurement was roughly 5%. The PNSD data from MPSS is also used for data quality checks with AMS + BC measurements. The mass closure between both measurements agrees with the correlation (r = 0.93) with a y = 1.11x + 1.06 slope, as shown in Fig. S13.

### 2.5 The ZSR mixing rule

The hygroscopicity parameter ($\kappa$) can be computed using the hygroscopic growth factor (GF) measured through an HTDMA, based on (Stokes & Robinson, 1966) and outlined by (Gysel et al., 2007).

$$\kappa_{measured} = (GF^3 - 1)\left(\frac{exp\left(\frac{A}{D_p dry \cdot GF}\right)}{RH} - 1\right), \tag{2}$$

$$A = \frac{4 \, \sigma s/a \, M_w}{RT\rho_w}, \tag{3}$$

$D_p$ dry and GF are the initial dry particle diameter and hygroscopic growth factor at 90 % RH measured by HTDMA. $\sigma s/a$ is the droplet surface tension (assumed to be that of pure water, $\sigma s/a = 0.0728$ N m$^{-2}$), $M_w$ the molecular weight of water, $\rho_w$ the density of liquid water, R the universal gas constant, T the absolute temperature. Alternatively, $\kappa_{chem}$ can be predicted using a simple mixing rule based on chemical volume fractions ($V_fi$), as proposed by (Petters and Kreidenweis, 2007):

$$\kappa_{chem} = \sum_i V_f i \cdot \kappa_i \tag{4}$$

Here, $\kappa_i$ and $V_fi$ are the hygroscopicity parameter and volume fraction for the individual (dry) component in the mixture, with i the number of components in the mixture. We derive $V_fi$ from particle chemical composition measured by AMS and AE33. The calculation of the volume fraction is described in detail in section 2.6. Throughout subsequent discussions, $\kappa_{measured}$ and $\kappa_{chem}$ denote the Kappa values derived from HTDMA and AMS plus AE33 measurements. The composition in this study indicates the presence of both hygroscopic and hydrophobic compounds, which may affect the particle's overall hygroscopicity. The approach we used to predict hygroscopicity follows the Zdanovskii-Stokes-Robinson (ZSR) mixing rule. To some extent, we indirectly incorporate solubility and chemical composition information in $\kappa$ calculations, particularly for particles with

significant fractions of organics more than 50%. Hence, in ZSR-based $\kappa$ predictions, the values of $\kappa_i$ used in this study (see Table 1) are based on literature-measured values, with some water activity calculation. Changing $\kappa_i$ in calculations doesn't significantly change $\kappa_{chem}$ prediction as lower soluble compounds like organics generally show only slight hygroscopic growth but a much better activity as a CCN than indicated by the hygroscopic growth, suggesting highly non-ideal behaviour for aerosol water contents at relative humidities less than 98% (Petters et al., 2009; Wex et al., 2009). The individual values of $\kappa_i$ reflect literature-measured and theoretical predictions as cited in Table 1. i.e., ammonium nitrate, ammonium sulphate, and organics are based on empirical measurements. Considering the uncertainties in the HTDMA and AMS measurements, the uncertainty between $\kappa_{measured}$ and $\kappa_{chem}$ of 10-15% (leading to an uncertainty of $\pm 0.06$ for $\kappa$) is roughly estimated, as shown in the supplementary.

**2.6 Hygroscopicity–chemical composition closure**

The Aerosol Mass Spectrometer (AMS) provides the particle mass concentrations, encompassing sulphate ($SO_4$), nitrate ($NO_3$), ammonium ($NH_4$) and chloride (Cl) ions, as well as the concentration of organic compounds. A simplified ion pairing scheme converts ion mass concentrations to the mass concentrations of their corresponding inorganic salts (Gysel et al., 2007).

**Table 1. Gravimetric densities ρ and hygroscopicity parameters κ were used in this study.**

| Species | $NH_4NO_3$ | $H_2SO_4$ | $NH_4HSO_4$ | $(NH_4)_2SO_4$ | Organic matter | Black carbon |
|---|---|---|---|---|---|---|
| $\rho$ (kg m$^{-3}$) | 1720[a] | 1830[a] | 1780[a] | 1769[a] | 1400[a, c] | 1770[a, b] |
| $\kappa_{measured}$ | 0.58[a] | 0.9[a, b] | 0.56[a] | 0.48[a] | $0.1 - 0.2$[a, c] | 0 |

[a](Gysel et al., 2007, 2011); [b](Park et al., 2004; Kondo et al., 2011; Wu et al., 2013); [c](Alfarra et al., 2006; Dinar et al., 2006)

**2.7 Air mass trajectory cluster analysis**

To determine air mass origin during specific pollution episodes, 72 h back trajectories were calculated on an hourly basis using the Hysplit trajectory with GDAS meteorological field data (SplitR package - RDocumentation, 2024) by (Stein et al., 2015). Back trajectories were set to end at SIRTA z5 coordinates (48.70° N, 2.14° E) at 100 m above ground level (a.g.l.). The clustering analysis provides insights into the influence of different source regions and atmospheric processes on local air quality. It improves understanding of aerosol transport mechanisms and their impact on the aerosol hygroscopic properties. The trajectory cluster analysis was performed using the Openair package from R (Rstudio, version 4.4.1), which uses the Angle method to get the cluster (Carslaw & Ropkins, 2012). The angle method of clustering is reliable and is beneficial for analysing the directional aspects of air mass transport, providing a better understanding of trajectory differences and enhancing the accuracy of clustering in trajectory analysis.

The optimal number of clusters was determined using the K-means elbow method, identifying five distinct clusters shown in Figure S3. Subsequent analysis classified hygroscopicity growth and other parameters based on these clusters. The clusters were designated as C2, primarily influenced by continental air masses; C1, influenced by

marine and continental air masses; and C3, C4, and C5 influenced by marine air masses traversing over land, explained in section 3.2.

## 3 Results and discussions

### 3.1 Hygroscopicity and Chemical Composition Overview

Figure 3. comprehensively represents the Growth Factor Probability Density Function (GF-PDF) and chemical composition throughout the measurement campaign. The sub-micrometre bulk particle chemical composition was derived from AMS and AE33 measurements. As shown in Figure S5, the meteorological conditions for the initial period correspond to heatwave-1 and local PM emissions, followed by a later period characterised by cloudy and rainy conditions, leading to a reduction in concentration. During measurements, the temperature reached 40°C

from the 17th to 19th of June, referring to heatwave-1, whereas wind speed was 1 to 5 ms$^{-1}$. The chemical composition during the entire measurement period indicates that organic matter accounted for approximately 60% of the total, with sulphate emerging as the second-largest contributor. The OM to OC ratio (Organic matter to Oxygen-Carbon) experiences a decline coinciding with an eBC rise, indicating the influence of black carbon in the composition, as illustrated in Figure S4, primarily associated with nearby burning plume event also seen in

the mass fraction during 28$^{th}$ June. Fig. 3a portrays the temporal evolution of AMS species and eBC. Elevated sulphate peaks observed on July 04$^{th}$ and 11$^{th}$ were attributed to transport events, either from industrial emissions in Rouen (Northern side of Paris) or shipping emissions in the Channel-Le Havre region, influenced by northwest winds. Throughout the campaign, transported air masses from the west, predominantly marine aerosols, were more influential, contributing to coarse aerosols such as sea spray and marine sulphates, as discussed in Section

3.2. The nitrate concentration was negligible during this measurement, so ammonium nitrate loss due to evaporation may not apply here. The coarse mode aerosol influence is also depicted in Figure S4, showing volume distribution from MPSS during the initial heatwave-1 period, which also coincides with high PM mass dominant with larger particles in volume. The GF-PDF illustrates a conspicuous size dependency and temporal variability in the hygroscopic growth of particles.

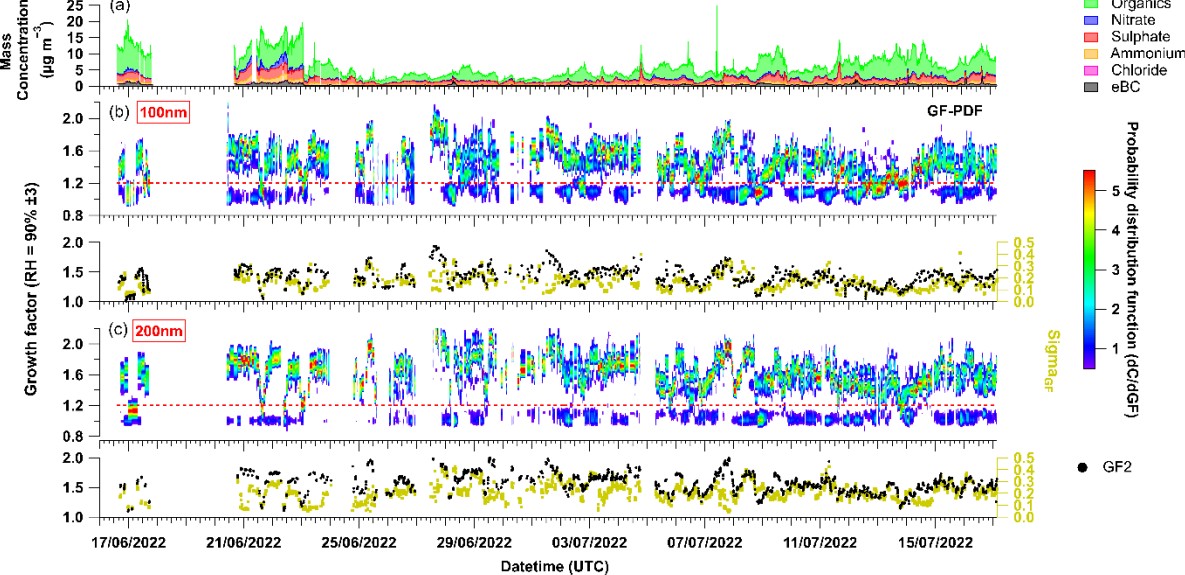

**Figure 3. Overview time series of chemical composition (a); size dependence of GF-PDF with GF2, which represents the mean growth factors of hygroscopic (GF > 1.2) particle and dotted red line for cut-off between hydrophobic and hygroscopic modes for size 100 and 200nm and sigma of GF ($\sigma_{GF}$) (b-c).**

In Fig. 3 b-c, the time series includes two distinct diameter sizes, namely 100 nm and 200 nm, selected to represent particles of small and large sizes. Comparable analyses have been applied to other diameters, as detailed in the supplementary materials. The GF-PDFs show two distinct modes: hydrophobic mode (GF < 1.2) and hygroscopic mode (GF > 1.2), also referred to as GF2 in Fig. 3b-c, signifying the prevalent mixing of aerosols. The colour scale represents the probability function (counts per GF), indicating a probability of a specific particle growth. A threshold value of GF 1.2 (red dotted line) is considered a cut-off line or differentiation between hydrophobic and hygroscopic particles for all sizes based on the mean of observed GF-PDF distribution where two modes are distinguished, represents a boundary where minimal water uptake occurs (Kim et al., 2020; Spitieri et al., 2023). This threshold GF coincides with the typical minimum in the observed GF-PDFs in this study, which also aligns with previous studies (Sjogren et al., 2008; Wu et al., 2013). However, occasionally, the number fraction of hygroscopic mode (F2 in Figure S6) approaches unity for particle sizes exceeding 100 nm. The hygroscopic mode consistently predominates in the GF-PDF across all particle sizes. With increasing particle size, the prominence of the hygroscopic mode intensifies, accompanied by a decline in the hydrophobic fraction (F1 in Figure S6). The hygroscopic mode (F2) mean number fractions are 0.75, 0.82, 0.84, and 0.84 for 100, 150, 200, and 250 nm particle sizes, respectively. It was hypothesised that larger particles had undergone atmospheric aging processes such as coagulation, condensation, chemical reaction, and cloud processing (Pöschl, 2005; Wu et al., 2013) for longer than smaller particles. These aging processes typically enhance the water solubility of particles (Pöschl, 2005; Jimenez et al., 2009; Wu et al., 2013). The $\sigma_{GF}$ can be derived from data inversion of HTDMA and can explain the degree of mixing of particles; the large $\sigma_{GF}$ tends to have multiple modes in GF distribution and wide growth spread, which is associated with external mixing and vice versa for internally mixed particles (Sjogren et al., 2008; Spitieri et al., 2023). In the time series, the high GF is also associated with a large $\sigma_{GF}$, which is prominently seen for larger diameters of 200nm. Larger $\sigma_{GF}$ values are primarily associated with a heterogeneous mixture of particles with diverse hygroscopic properties. In some cases, such as on 29[th] June and 08[th] July, smaller particles grow as organics mass fraction increases, but no significant change in GF is observed, as shown in PNSD, Figure S4. This is hypothesised to be related to a new particle formation that needs further investigation, as this is not the focus of this study. Correlating the volume fraction of chemical species with hygroscopicity will give a more general trend of the contribution of each chemical component to overall hygroscopic properties. It can quantify the variation of the water uptake capacity of aerosol.

### 3.1.1 Correlation between measured hygroscopicity, organic and inorganic volume fraction.

The correlation between measured hygroscopicity (GF) and the organic volume fraction is illustrated in Fig. 4a, revealing a negative correlation between the two variables. The anti-correlation is evident as the hygroscopicity decreases with an increasing volume fraction of organics, which makes organic less hygroscopic, which also aligns with a previous study by (Kamilli et al., 2014). The linear regression in Fig. 4a is evaluated as y = -0.57x + 0.2. Fig. 4b shows the correlation of the volume fraction of the inorganic compounds with GF during the campaign. Here, inorganic compounds are the sum of all species from AMS measurement except organics, i.e., the sum of nitrate, sulphate, ammonium and chloride. In contrast to the organic fraction, the inorganic compounds have a noticeable promoting effect on hygroscopic growth. The time series of the inorganic volume fraction and

GF follow a similar pattern but with differing magnitudes with a correlation (r = 0.56). Here, the outliers are the high hygroscopic events during the campaign period, which may influence the correlation.

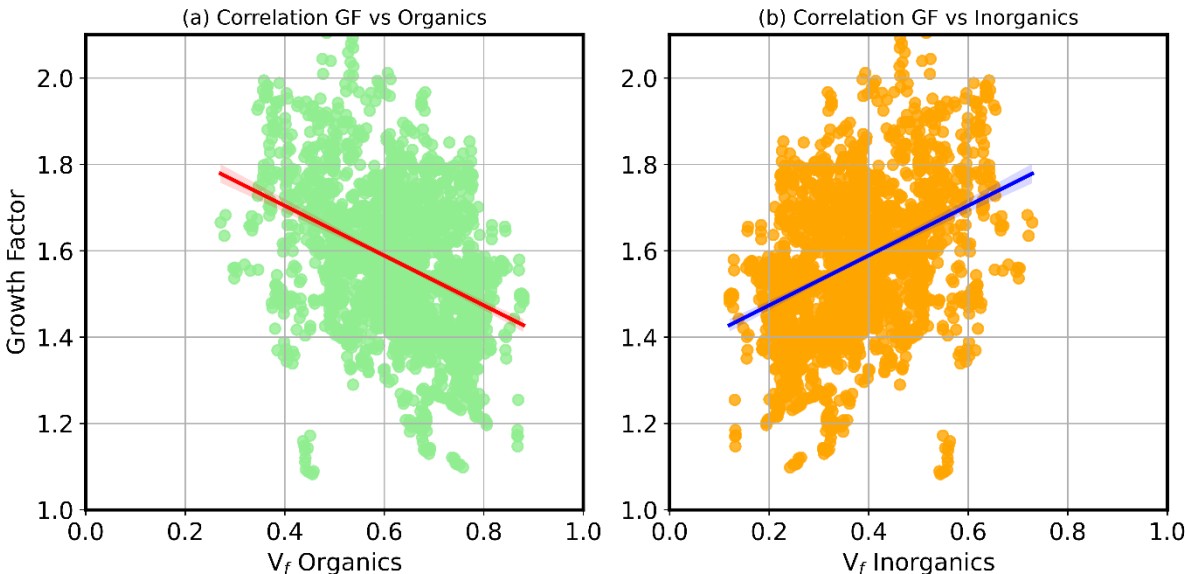

**Figure 4. Comparison between measured GF and volume fraction ($V_f$) of organics and inorganics for 200nm.**

The volume fractions ($V_f$) and their correlation with GF for different particle sizes help in understanding how the balance of organic and inorganic materials influences overall hygroscopic behaviour. $V_f$ shows their relevance for interpreting trends in measured GF and their application in $\kappa_{chem}$ calculations, where they are explicitly used to represent the contribution of different chemical species. This linkage helps connect the measured hygroscopicity with chemical composition and highlights the importance of volume fractions in understanding the discrepancies

between measured and predicted hygroscopicity. It can be further used in closure and trajectory cluster analysis.

**3.2 Hygroscopic Closure using PM1 Bulk Chemical Composition and Trajectory Cluster Analysis**

In this section, hygroscopicity parameters are denoted as $\kappa_{measured}$ and $\kappa_{chem}$. Corresponding $\kappa_{chem}$ values are calculated using the ZSR mixing rule based on chemical composition. In principle, the values of both approaches agree quantitatively and are within the range of their uncertainty, and then closure is achieved (Gysel et al., 2007;

Wu et al., 2013).

$\kappa$ is evaluated using equations 2 and 4, representing measured and chemically derived values at 90% RH. The variation of kappa over time is depicted in Fig 5, primarily influenced by particle size and chemical composition. The $\kappa_{measured}$ values for smaller particle sizes, specifically 100 nm and 150 nm, range between 0.24 and 0.29, as outlined in Table S1. These values increase to 0.35 for larger sizes, particularly for accumulation mode particles

of 200 nm. In Fig 5. box plots show the dataset overview and uncertainty with a shaded region as a standard error on the line plot. It should be noted that the uncertainty involving instrument measurement and different corrections of the dataset (i.e., RH accuracy for HTDMA or volume fraction of different AMS species) was already explained in Section 2 and sub-sections. The mean $\kappa$ for each dataset with standard error as uncertainty is shown with the error bar in Fig S12. It is hypothesised that the high hygroscopicity of accumulation mode particles is attributed

to their comparatively larger surface area concentration and inherent water-absorbing capabilities (Liu et al., 2014).

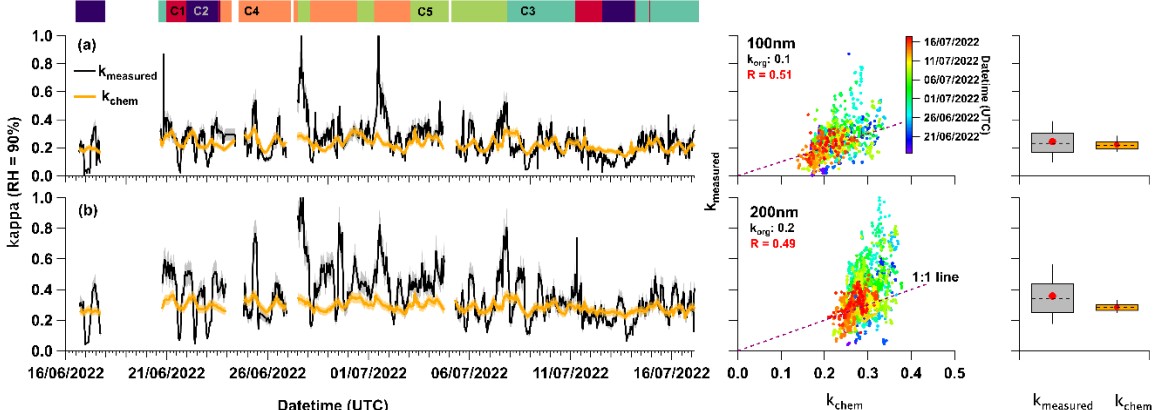

**Figure 5. a-b) The time series and comparison of κ_measured and κ_chem with the correlation plot for 100 and 200nm and clusters classification over time at the top. The box plots represent the hygroscopic parameter kappa (κ) of respective sizes of particles for measured and chemically derived in which low and high whisker traces represent the 9 and 91 percentiles, respectively. The red marker indicates the average of the data, whereas the upper and lower sides of the boxes indicate the 75 and 25 percentiles of the data, respectively.**

In most cases, κ_chem closely mirrors the trends observed in the directly κ_measured, barring a few outliers or instances of elevated hygroscopic peaks. These anomalies are occasionally linked to transported air masses from the northwest or southwest, potentially influenced by sea spray or highly hygroscopic salts non-detectable by AMS, and further compounded by the lack of on-site measurements during the campaign. The notable hygroscopic peaks observed from late June to the first week of July are tentatively attributed to transported airmass carrying aerosols from western to southwest, as indicated by HYSPLIT trajectory cluster analyses. We speculated that this is related to marine airmass with high hygroscopic coarse aerosol particles like sea salt (NaCl) and sulphate salt or dimethyl sulphide (DMS). Marine air masses typically exhibit higher hygroscopicity than continental air masses (Huang et al., 2022).

The closure between measured and chemical-derived hygroscopicity for particles of 100 nm corresponding to the Aitken mode or smaller sizes almost overlaps except during high peak events, correlating at approximately 51%, as depicted in Fig. 5a. This congruence is not observed for larger sizes, which is speculated to arise from the uncertainty associated with the κ_chem value for externally mixed particles and the bimodal distribution of aerosols. The frequency of the bimodal distribution increases with particle size. Data segregation based on wind speed, direction, and high hygroscopicity did not make significant differences in correlation; the larger sizes correlation declined, as illustrated in Fig. S8. The κ_org in chemically-derived hygroscopicity is considered 0.1 and 0.2 for small and large particle sizes, respectively, as theoretically inspected by overlapping with measured hygroscopicity. κ_org = 0.2 for larger sizes of 200 and 250nm is based on the slightly better correlation between κ_chem and κ_measured with the overlapping trend for time series. The κ_org parameter represents the hygroscopicity of organic matter in aerosols. This parameter can vary (e.g., 0.1 to 0.2) based on factors like the type of organic compounds present. Some organic compounds may be more hygroscopic, corresponding to higher κ_org values with a more oxygenated organic aerosol (OOA), while others may be less hygroscopic (Chang et al., 2010; Gunthe et al., 2011; Jimenez et al., 2009b; Petters and Kreidenweis, 2007). The nature of the organic matter and its interaction with water is based on factors like composition and environmental conditions, which are reflected in the difference between κ_org in ZSR calculations. The outliers in the κ_measured period reflect poor correlation, prompting trajectory analysis.

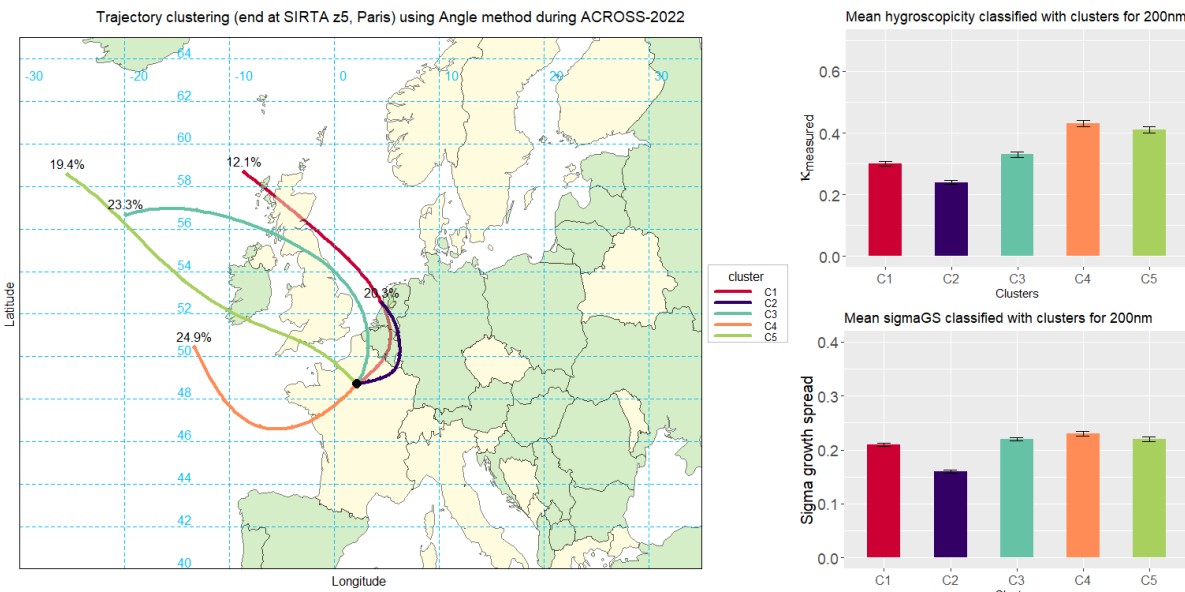

**Figure 6. a) Trajectory cluster for the campaign period, b) Mean bar plot of $\kappa_{measured}$ and $\sigma_{GF}$ spread classified with clusters for 200nm.**

Trajectory cluster analysis was conducted to investigate the high hygroscopicity events where discrepancies between $\kappa_{chem}$ and $\kappa_{measured}$ values were most pronounced. As depicted in Fig. 6a, five distinct trajectory clusters were identified: C1, C2, C3, C4, and C5. All clusters except C2 were generally influenced by marine air masses containing coarse particles, such as sea salt and dimethyl sulphide (DMS). Cluster C1, contributing 12.1% of the trajectories, passed over land-sea regions, suggesting that these air masses spent significant time over continental areas, thereby introducing local pollutants mixed with coarse marine particles to the measurement site. Cluster C2, which contributed 20.3% of the air mass, remained over the continent throughout its trajectory, originating from Northern West France, Belgium, and the Netherlands. This air mass passed over industrial regions, carrying local pollutants.

Clusters C3, C4, and C5 contributed 67.6% of the trajectories, predominantly from marine regions. Specifically, C4 trajectories came from the southwest, with the air mass traversing southern France, likely transporting a mix of biogenic and oceanic aerosols to the site. Trajectories in C5 spent minimal time over the continent, carrying predominantly marine air masses and coarse particles. The parameters were classified according to the identified clusters to gain further insights into the variations in hygroscopicity and $\sigma_{GF}$, as illustrated in Figure S9. These time series plots, presented as stacked bar charts, clearly show that high hygroscopic values were associated with C4 and C5 trajectories, which were influenced by coarse marine particles. Correspondingly, the $\sigma_{GF}$ was also high for these trajectories, as seen in the mean values from the bar plot Fig 6b for 200nm.

Notably, during a high GF event on June 27, 2022, the air mass was primarily from C5, and a significant difference was observed between $\kappa_{chem}$ and $\kappa_{measured}$. This discrepancy is likely due to the limitations of the AMS in measuring coarse or sea salt particles. Additionally, the lack of sea spray information in the chemical composition and the ZSR mixing rule prevented accurate prediction of hygroscopicity by $\kappa_{chem}$. Furthermore, the $\sigma_{GF}$ during this event was 0.3, indicating the presence of multiple modes and externally mixed particles, which the ZSR mixing rule cannot account for.

Towards the end of the campaign, from July 11 to July 17, 2022, the air masses were predominantly influenced by C1, C2, and C3, with a more substantial influence from local air masses, pollutants, and eBC. Since eBC is

hydrophobic and has a low GF, the agreement between $\kappa_{chem}$ and $\kappa_{measured}$ during this period was close to the 1:1 line in Fig 5b. In the ZSR mixing model, eBC is assumed to have a kappa value of zero, and the $\kappa_{measured}$ during this period was around 0.4, indicating minimal influence from coarse particles on the predictions. The $\sigma_{GF}$ for 100 nm particles during this time was less than 0.2, as shown in Fig. S9. This partially explains the stronger correlation between $\kappa_{chem}$ and $\kappa_{measured}$ for smaller diameters (e.g., 100nm) compared to larger particle diameters (e.g., 200 – 250nm). The presence of multiple modes and externally mixed particles change the characteristics of aerosol and influence the prediction of hygroscopicity. To show the influence of the mixing state on hygroscopicity, the following analysis is described in section 3.3.

### 3.3 Variability in GF-PDFs and Particle Mixing State Influence

#### 3.3.1 GF-PDF Analysis from the ACROSS Campaign

As depicted in Fig. 7, the mean GF-PDFs were computed by averaging individual GF-PDFs for each particle size with $D_0$ of 100, 150, 200 and 250 nm over the entire campaign period. The mean GF-PDFs offer insights into the growth distribution, but they may not provide a clear depiction of the mixing state within these size fractions. However, the $\sigma_{GF}$, explained in earlier sections and seen as the widespread distribution of GF 1 – 2.2 on the x-axis in Fig 7, can give insights into the mixing state.

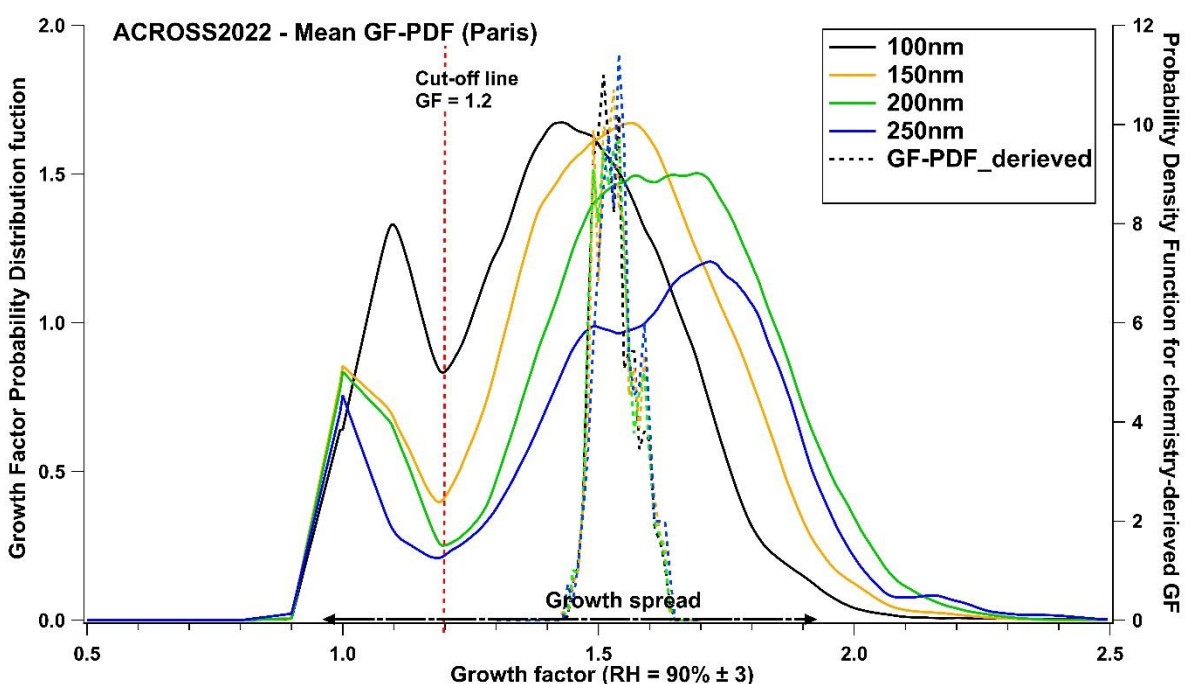

**Figure 7. The mean GF-PDFs for different diameters and the vertical red line represent the selected cut-off between the hydrophobic (GF < 1. 2) and the hygroscopic mode (GF > 1. 2).**

The predicted GF-PDF is calculated using the corrected growth equation (2) used in the paper by (Sjogren et al., 2008), also mentioned in the supplementary. First, the calculated $\kappa_{chem}$ from the ZSR mixing rule was used, and then the corresponding corrected growth at 90% was calculated using the equation. After converting $\kappa_{chem}$ to $GF_{chem}$, the frequency distribution and the probability density function were calculated. This frequency distribution and probability function is calculated using the Microsoft Excel data analysis tool. The predicted GF-PDF is represented as dashed lines and the measured GF-PDF as solid lines in Fig 7. The mean measured GF-PDF shows

a bimodal to multi-modal distribution with a wide growth spread exhibiting dominance of external mixing during the measurement period. In contrast, predicted GF-PDF derived from $\kappa_{chem}$ presents a narrower growth spread and a mono-modal distribution with GF 1.4 – 1.7. Whereas the chemical composition from AMS is bulk without coarse particle information, it incorporates ZSR-based bulk aerosol hygroscopicity predictions that provide mean

growth values exclusively and do not facilitate the identification of distinct growth modes, instead treating the ensemble as a perfect internal mixture. The measured GF-PDF incorporates all aerosol populations, giving size-resolved GF and growth distribution. This can also influence the closure agreement. Furthermore, during clean conditions, slight variations in the average bulk inorganic ion mass fraction at a given diameter may lead to significant changes in the external mixing of the particle population. While this exerts a minor influence on the

mean growth value, it significantly impacts the associated GF-PDF (Healy et al., 2014). A similar approach was performed when the correlation between $\kappa_{chem}$ and $\kappa_{measured}$ was better in the later part of the campaign from the 13[th] to the 17[th] of July. However, no significant change was seen in GF-PDFs except for a slightly narrower growth spread of measured GF-PDF. To understand and get more insights into the mixing state and hygroscopicity of different diameters during the fresh emission, the diurnal pattern was compared with eBC properties. A similar

comparison was also performed with data available on the measurement sites for other gases.

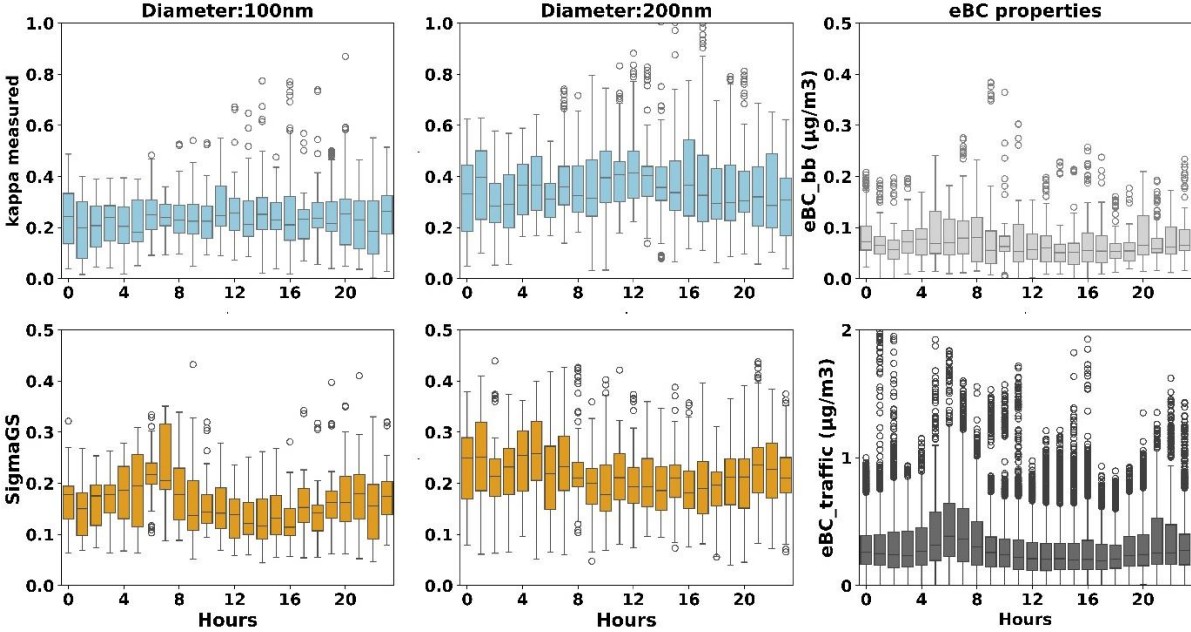

**Figure 8. Diurnal Percentiles of the $\kappa_{measured}$ and $\sigma_{GF}$ spread for 100 and 200nm with eBC properties.**

Figure 8 compares the diurnal patterns of $\kappa_{measured}$, $\sigma_{GF}$ and eBC properties. The discrimination of black carbon (eBC) into biomass burning (eBC_bb) and traffic fractions, employing the so-called "aethalometer model"

(Drinovec et al., 2015; Sandradewi et al., 2008; Srivastava et al., 2019) revealed distinct behaviours about particle size. The eBC traffic exhibited a morning rush hour peak around 0600 hrs (Zhang et al., 2019), coinciding with a spike in $\sigma_{GF}$, suggesting an external mixing of particles with eBC. Furthermore, $\sigma_{GF}$ follows the eBC traffic diurnal variation trend for 100nm. In contrast, the $\kappa_{measured}$ exhibits an opposing trend, aligning with the non-hygroscopic nature of eBC and its contribution to the particle mixture morphology. No significant trend relation was observed

among the three parameters for accumulation mode particles of 200nm. The contribution of the hygroscopic mode to total hygroscopicity exceeded that of 100 nm particles. Larger particles, typically more aged than smaller

counterparts, with higher hygroscopic growth (Cubison et al., 2006), are more closely associated with atmospheric processing during long-range transport (Kalivitis et al., 2015; Spitieri et al., 2023). This could explain further the 51% agreement in closure for 100 nm particles, where mixing and eBC tend to be more homogeneous and limited
to a bimodal distribution, unlike multiple modes for larger sizes.

### 3.3.2 Comparison with other measurement data

Another investigation was performed on a different dataset to assess further and validate the closure study conducted during the ACROSS campaign. The data from the measurement campaign, which took place in Thuringia, central Germany, spanning from September to October 2010, as part of the Hill Cap Cloud Thuringia
(HCCT) at the Schmücke research site. The results and description of this study have been previously published by (Wu et al., 2013). Their closure study reported a better correlation between predicted and measured hygroscopicity, yielding correlation coefficients of 0.66 and 0.44 for larger and smaller particles, respectively. Analysis of similar measured and predicted GF-PDF is presented in Fig. S10. Notably, the mean measured GF-PDF exhibit a narrower growth spread and distinct modes, which overlap with predicted GF-PDF. The comparable
growth spread for measured and predicted suggests that aerosol particles during the HCCT campaign were aged and represented a well-distinguished mixture of compounds. Taking note, Schmücke is a pristine and mountainous region with not much variation in aerosol population except transportation events, and the measurement period was during winters, which will further support the better correlation of closure. As a consequence of our study, the influence of the mixing state in the prediction of hygroscopicity is seen. Further similar analysis should be
done for different datasets to get insights into particulate mixing influence and provide a threshold value for which the $\kappa_{chem}$ derived using the classical ZSR mixing rule cannot be used. Due to the lack or unavailability of another dataset from Paris for comparison, the threshold value for $\kappa_{chem}$ cannot be improved.

The mean GF-PDF and chemical composition during the ACROSS campaign suggest the air mass was a mixture of complex compounds and externally mixed particles. Occasionally, aerosol source emissions are in close
proximity. The external mixing is critical and has a complex composition of atmospheric aerosols. The role of the aerosol mixing state in determining climate-relevant properties such as CCN activity and aerosol optical characteristics (Riemer et al., 2019; Wang et al., 2020) needs advanced measurement techniques and modelling approaches, which may be computationally time-consuming and expensive. Using GF-PDFs to analyse aerosol mixing states adds a new perspective of understanding how internal and external mixtures impact hygroscopic
predictions. This complexity could contribute to the limited agreement in the closure study. Most referred studies use the CCN measurements to derive $\kappa$, but mixing information is lacking in these measurements, whereas external mixing using growth spread can improve the understanding of hygroscopicity. The internally and externally mixed particles exhibit different water-uptake behaviours and the importance of considering it in predicting aerosol hygroscopicity and suggested that using multiple complementary measurement techniques can provide a more
comprehensive understanding of aerosol water-uptake properties (Razafindrambinina et al., 2022). Our findings highlight the importance of external mixing, and an approach to growth spread in hygroscopicity can improve the modelling parameters. The inadequacy of the ZSR mixing rule is evident in polluted and mixed aerosol environments, challenging the classical method's applicability for hygroscopicity comparison and prediction. One must consider many assumptions to make it work. Similar studies should be needed to evaluate more knowledge
about externally mixed particles and chemical-derived hygroscopicity based on the classical ZSR mixing rule at

different polluted environments and sources. It highlights the limitations of the ZSR mixing rule in predicting hygroscopic growth for externally mixed aerosols, which are common in urban to sub-urban areas affected by fresh emissions and originate from various sources, including long-range transport and secondary organic aerosols. This approach gives a spectrum of size-resolved aerosol hygroscopic behaviour beyond bulk chemical composition. It can enhance the characterization of aerosol hygroscopicity in climate models, as it directly influences the cloud microphysical processes and radiative forcing. This method can improve the accuracy of climate model simulations.

**Conclusion**

Aerosol hygroscopicity has been investigated for the peri-urban site in Paris using growth factor measurements by HTDMA at a relative humidity of 90 %. Additionally, the ZSR mixing rule was used to predict hygroscopic growth factors based on the aerosol chemical composition measured by AMS and AE33. The results indicate a pronounced size dependence of hygroscopic growth. During the measurement period, the median $\kappa$ values of 100, 150, 200, and 250 nm particles derived from the HTDMA measurements are 0.24, 0.29, 0.35, and 0.36, respectively. Organics constitute approximately 60% of the total chemical composition during the measurement period, with sulphate as the second-largest contributor at 20%. The aerosol's mixing state is characterised based on the standard deviation or sigma ($\sigma$) of the inverted GF-PDF, distinguishing between internal ($\sigma_{GF} < 0.08$) and external ($\sigma_{GF} > 0.10$) mixing. The correlation between chemical-derived and measured hygroscopicity is 51% for smaller-size particles, and the correlation declines for larger sizes. The temporal evolution of the predicted growth factor does not coincide well with the measured growth factor. Integrating chemical composition data with back trajectory clustering offers a novel approach to understanding how different air masses contribute to discrepancies between $\kappa_{measured}$ and $\kappa_{chem}$ values, particularly during high hygroscopic events. Possible reasons for the limited agreement between measured and derived growth factors are:

- Differences in GF-PDFs and growth spread.
- Limitations of ZSR-based bulk aerosol hygroscopicity predictions provide mean GF values exclusively but do not distinguish GF modes.
- Assigned compounds where organics input is not well-defined and incorporated in ZSR mixing rule computations with inaccuracy.
- Insufficient ambiguous ion balance calculations and information on inorganic salts from marine airmass and size segregate chemical composition. Size segregation can slightly improve the correlation.

However, it should be noted that the predicted GFs underestimate the measured GFs. These findings emphasise the need for caution when using such chemical-derived hygroscopicity measurements in climate models, especially in regions with fresh aerosol emissions close to the source. The current results of combining hygroscopicity measurements with trajectory cluster analysis and $\kappa_{chem}$ predictions represent a significant value in hygroscopicity prediction. Our findings align with previous studies highlighting the significant influence of the mixing state on aerosol hygroscopicity. Most referred studies use the CCN measurements to derive $\kappa$, but mixing information is lacking in these measurements. In contrast, external mixing using growth spread, especially in a sub-saturated regime, can improve the understanding of particle mixing in hygroscopicity prediction. The particle mixing significantly impacts global climate prediction, mainly based on local and regional models. Additionally,

external mixing hygroscopicity is likely to impact aerosol-related health effects, such as lung deposition in a

populated area where people typically live. These improve understanding of aerosol growth in sub-saturated conditions and their broader impact on atmospheric processes, including aerosol microstructure and optical properties. The HTDMA data, however, can be further parameterised and utilised in machine learning to predict hygroscopicity using different models. It can also be used as a proxy for predicting Cloud Condensation Nuclei (CCN) or $\kappa$ predictions.

**Code availability**

The Trajectory clustering code and other processing toolkit can be made available upon request to the first author (deshmukh@tropos.de).

**Data availability**

Level 2 datasets used in the present study from the ACROSS field campaign for the SIRTA site are available or

will be made soon available on the AERIS data center (https://across.aeris−data.fr/catalogue/) and DOI referenced. Also, data are available upon request to the author (deshmukh@tropos.de).

**Author contribution**

SD performed the formal analysis and wrote the original draft. SD and LP performed the investigation and data curation. JEP, LP, SD and OF coordinated the ACROSS field campaign. SD, LP, PF, JEP, and OF contributed to

the campaign setup, deployment, calibration, and operation of the instrumentation, as well as the data collection and analysis from the SIRTA site. SD, LP, MP and SH provided the methodology and conceptualisation. MP and HH provided supervision and validation. LP, BW, JEP and SH contributed to reviewing and editing the manuscript. All authors commented and contributed.

**Competing interests**

The contact author has declared that none of the authors has any competing interests.

**Disclaimer**

Publisher's note: Copernicus Publications remains neutral concerning jurisdictional claims in published maps and institutional affiliations.

**Funding**

This work has been supported by the DFG H-AMS project with the reference WE 2757/4-1 and is involved in the ACROSS campaign. The ACROSS project has received funding from the French National Research Agency 475 (ANR) under the investment program integrated into France 2030, with the reference ANR−17−MPGA−0002,

and it was supported by the French National program LEFE (Les Enveloppes Fluides et 610 l'Environnement) of the CNRS−INSU (Centre National de la Recherche Scientifique/Institut National des Sciences de l'Univers).

**Acknowledgements**

We acknowledge support from the SIRTA facility in conducting measurements and collaboration. Support from other groups involved in measurement with us from LSCE (Laboratoire des Sciences du Climat et de l'Environnement), LAMP (Laboratoire de Météorologie Physique), Etienne Brugere for supporting in measurements and Institut National de l'Environnement Industriel et des Risques members is appreciable. I want
to acknowledge support from TROPOS, University of Leipzig and PhD colleagues.

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
