# Peer review of "External particle mixing influences hygroscopicity in a suburban area"

_EGUsphere, 2024_

## Author Response (AR1)

Firstly, we would like to thank the reviewers for carefully reading the paper and providing valuable comments that helped improve the quality of the manuscript. We have considered all the comments raised by the reviewers and have changed the paper accordingly. The details of our changes are highlighted in the text and color-coded. The point-by-point answers to Reviewers #1 and #2 are provided below.

**Color code:** Black: comment from reviewer; Blue: Reply/Response and sentence from the manuscript; Brown: Change/addition in the revised manuscript.

**Response to Referee comments #RC1**

**[General Comment]** The paper is generally well written and the figures support the key findings. I recommend publication after resolving minor concerns.

**Response:** Thank you very much for your comment.

Minor concerns:

**[General Comment]** Sec. 2.3 The authors introduce particle size distributions, but neither show them in the paper nor in the supplemental material. If the authors did not find any distinct size distributions across air masses or mixing ratios, it is still worth expressing that.

**Response:** Thank you,

PNSD measurements analysis is considered during the campaign and for AMS mass concentration quality checks. This was not mentioned in the text earlier. We added a sentence in section 2.4 and a description in the result section 3.1. The QC plot is now shown in the supplementary (Fig. S13).

[Ln 189 – 190] The PNSD data from MPSS is also used for data quality checks with AMS + BC measurements. The mass closure between both measurements agrees with the correlation (r = 0.93) with a y = 1.11x + 1.06 slope, as shown in Fig. S13.

[Ln 268-270] The coarse mode aerosol influence is also depicted in Figure S4, showing volume distribution from MPSS during the initial heatwave-1 period, which also coincides with high PM mass dominant with larger particles in volume. The GF-PDF illustrates a conspicuous size dependency and temporal variability in the hygroscopic growth of particles.

[Ln 294-296] In some cases, such as on 29th June and 08th July, smaller particles grow as organics mass fraction increases, but no significant change in GF is observed, as shown in PNSD, Figure S4. This is hypothesised to be related to a new particle formation that needs further investigation, as this is not the focus of this study.

[Figure]

Figure S4. The time series of PNSD and volume distribution with Kappa and eBC.

[Figure]

Figure S13. The mass closure for data quality checks with MPSS and AMS + BC mass concentrations.

**Supplementary:** Total mass concentrations (AMS+BC) are calculated, and density correction is applied with the effective density ($\rho$) of individual species (Org, NO3, SO4, NH4, Cl and BC) from literature (Park et al., 2004; Kondo et al., 2011; Poulain et al., 2014). The agreement between mass concentration from both instruments is correlated (r = 0.93) with a slope of y = 1.11x + 1.06, which is suitable for data quality.

**[General Comment]** Fig. 1 Please explain the upward pointing arrows. Assuming these are sources, are there any sinks worth highlighting?

**Response:** The background is adapted from the ACROSS campaign descriptor (cited in figure caption) https://across.aeris-data.fr/description/. We have explained it in the text now (revised manuscript version). There are no sinks worth highlighting in the figure for this campaign context.

[Figure]

[Pg3, Ln 96-100] One month of measurements during the ACROSS campaign provides insights into microphysical parameters, including time and size-resolved HTDMA data under sub-saturated conditions in a suburban environment, shown in Fig. 1. The upward arrows define the processes by which primary pollutants, such as oxides of nitrogen and volatile organic compounds in red, pink and blue colours, respectively, are emitted in the atmosphere, leading to their oxidation and ultimate removal while at the same time producing secondary species, such as ozone and organic aerosols in green colours over suburban to forest region.

**[General Comment]** Fig. 5 Please indicate the uncertainty, for example, by showing a typical error bar per data set (e.g., on the far left or far right).

Response: We revised the figure with box plots to show the dataset overview and uncertainty with the shaded error region on the line plot. It is also mentioned in the caption and the text.

[Figure]

Figure 5. a-b) The time series and comparison of $\kappa_{measured}$ and $\kappa_{chem}$ with the correlation plot for 100 and 200nm and clusters classification over time at the top. The box plots represent the hygroscopic parameter kappa ($\kappa$) of respective sizes of particles for measured and chemically derived in which low and high whisker traces represent the 9 and 91 percentiles, respectively. The red marker indicates the average of the data, whereas the upper and lower sides of the boxes indicate the 75 and 25 percentiles of the data, respectively.

[Pg11, Ln 342-349] In Fig 5. box plots show the dataset overview and uncertainty with a shaded region as standard error on the line plot. It should be noted that the uncertainty involving

instrument measurement and different corrections of the dataset (i.e., RH accuracy for HTDMA or volume fraction of different AMS species) was already explained in Section 2 and sub-sections. The mean κ for each dataset with standard error as uncertainty is shown with the error bar in Fig S12. It is hypothesised that the high hygroscopicity of accumulation mode particles is attributed to their comparatively larger surface area concentration and inherent water-absorbing capabilities (Liu et al., 2014).

[Figure]

Figure S12. The mean kappa with an error bar to show the uncertainty for each dataset for 100 & 200nm

**[General Comment]** l. 251-254 (and throughout) Please be explicit about material that is being discussed (as opposed to results). The authors could use distinct phrases, such as "We speculate", etc.

**Response:** We revised the sentence and also go through the whole manuscript:

[Pg8, Ln299] It was hypothesised that larger particles had undergone atmospheric aging processes such as coagulation, condensation, chemical reaction, and cloud processing (Pöschl, 2005; Wu et al., 2013) for longer than smaller particles. These aging processes typically enhance the water solubility of particles (Pöschl, 2005; Jimenez et al., 2009; Wu et al., 2013).

[Pg10, Ln 350] It is hypothesised that the high hygroscopicity of accumulation mode particles is attributed to their comparatively larger surface area concentration and inherent water-absorbing capabilities (Liu et al., 2014).

[Pg11, Ln 369] We speculated that this is related to marine airmass with high hygroscopic coarse aerosol particles like sea salt (NaCl) and sulphate salt or dimethyl sulphide (DMS). Marine air masses typically exhibit higher hygroscopicity than continental air masses (Huang et al., 2022).

**General Comment]** l. 262 Please define "inorganics" upon first use. Is it simply the sum of all species except "organics"?

Response: We revised the sentence and added the following.

[Pg9, Ln319-320] Figs. 4b shows the correlation of the volume fraction of the inorganic compounds with GF during the campaign. Here, inorganic compounds are the sum of all species from AMS measurement except organics, i.e., the sum of nitrate, sulphate, ammonium

and chloride. In contrast to the organic fraction, the inorganic compounds have a noticeable promoting effect on hygroscopic growth.

**General Comment]** l. 270 A correlation of 0.56 does not seems that high, leaving ~68% of the variance unexplained. Using "reasonable" seems confusing here.

Response: We revised the sentence and avoided this statement.

[Pg9, Ln322] The time series of the inorganic volume fraction and GF follow a similar pattern but with differing magnitudes with a correlation (r = 0.56). Here, the outliers are the high hygroscopic events during the campaign period, which may influence the correlation.

**General Comment]** ll. 274-275 It is unclear whether the authors motivate the next section here or whether they propose work that won't be covered in this paper. Please clarify.

Response: We revised the sentence. This sentence explains the volume fraction ($V_f$) and its relation with GF, which must be taken into account for a rough estimation of trends with organic and inorganics. Whereas in chemical-derived kappa calculation, $V_f$ is used explicitly.

[Pg9, Ln316] The anti-correlation is evident as the hygroscopicity decreases with an increasing volume fraction of organics, which makes organic less hygroscopic, which also aligns with a previous study by (Kamilli et al., 2014).

[Pg 10, Ln 332 – 336] The volume fractions ($V_f$) and their correlation with GF for different particle sizes help understand how the balance of organic and inorganic materials influences overall hygroscopic behaviour. $V_f$ shows their relevance for interpreting trends in measured GF and their application in $\kappa_{chem}$ calculations, where they are explicitly used to represent the contribution of different chemical species. This linkage helps connect the measured hygroscopicity with chemical composition and highlights the importance of volume fractions in understanding the discrepancies between measured and predicted hygroscopicity. It can be further used in closure and trajectory cluster analysis.

**General Comment]** ll. 294-296 It appears as though the authors are suggesting high hygroscopicity pollutants here. Please clarify and, if true, perhaps introduce in Section 1.

Response: Thanks for the comment. Sorry for the mistake. We were suggesting the high hygroscopic peaks due to transported airmass carrying aerosols, and this is assumed mainly related to sulphate or sea spray (highly hygroscopic), which is also shortly explained in section 3.1 [Pg 8, Ln 259 - 267]. We revised the sentence and changed it.

[Pg11, Ln 369-373] The notable hygroscopic peaks observed from late June to the first week of July are tentatively attributed to transported airmass carrying aerosols from western to southwest, as indicated by HYSPLIT trajectory cluster analyses. We speculated that this is related to marine airmass with high hygroscopic coarse aerosol particles like sea salt (NaCl) and sulphate salt or dimethyl sulphide (DMS). Marine air masses typically exhibit higher hygroscopicity than continental air masses (Huang et al., 2022).

Section 3.1 Ln 272 Throughout the campaign, transported air masses from the west, predominantly marine, were more influential, contributing to coarse-mode aerosols such as sea spray and marine sulphates, as discussed in Section 3.2.

**[General Comment]** Minor grammatical and typo improvements:

Line 104: "The k will examine" seems an odd construct. Perhaps substitute with "k will inform".

Response: We revised the sentence to 'k will inform the relationship between the measured and the chemical-derived hygroscopicity.' [Pg3, Ln108]

Line 116 and throughout the manuscript: Please remove parentheses within parentheses.

**Response**: Thanks; we revised and changed them throughout the manuscript. [Pg3, Ln115; Pg7, Ln236]

Line 396: Perhaps replace "another" with "other"

**Response:** We revised the sentence and changed it to. [Pg15, Ln478]

3.3.2 Comparison with other measurement data

**Response to Referee comments #RC2**

**[General Comment]** The paper is a good application of measurements to identify important considerations (mixing state, chemical composition) in predicting hygroscopicity in subsaturated conditions. I recommend publication after addressing some minor comments.

**Response:** Thank you very much for the comments.

Minor concerns:

**[General Comment]** Page 5, Line 141: Does a variance of +/- 3% RH have a significant effect on GF measurements in this study? GF especially for lower soluble compounds may vary significantly within the +/- 3% RH range. If there is an effect, I would recommend mentioning this as a source of uncertainty.

**Response:** Thank you for the comment; we revised and mentioned it in the manuscript. For this HTDMA and from the calibration shown in Fig. S2, GF can differ by less than ±0.1 for Ammonium Sulphate and lower soluble compounds. Also, the E-AIM model by S. Clegg shows that the variation in GF for less soluble organics is negligible below 90% RH. Although the RH calibration was maintained carefully, this uncertainty is acknowledged as a potential source of error in our measurements.

[Pg 5, Ln 141-145] The selected dry diameters $D_0$ for this campaign at specific narrow size fractions centred around 100, 150, 200 and 250 nm in a differential mobility analyser (DMA-1) and exposed at relative humidity (RH) of $90 \pm 3$ % (Bezantakos et al., 2013; Wu et al., 2013).

Both DMAs were operated with a sheath flow rate of 5 l min−1 and a sample flow rate of ~1 l min−1. It is to be noted that the variance of RH may have a measurable impact on GF, particularly with low-soluble compounds and their sensitivity to RH in this range. This could contribute to GF values uncertainty, especially organics or less-hygroscopic dominant species. Although the RH calibration was maintained carefully, this uncertainty is acknowledged as a potential source of error in our measurements.

**[General Comment]** Section 2.3: Provide more information/description regarding the particle size distribution in the results as it is introduced in the methodology

**Response:** Thank you. I have **already answered #RC1** and added more description regarding PNSD to the result.

PNSD measurements analysis is considered during the campaign and for AMS mass concentration quality checks. This was not mentioned in the text earlier. We added a sentence in section 2.4 and a description in the result section 3.1. The QC plot is now shown in the supplementary (Fig. S13).

[Ln 189 – 190] The PNSD data from MPSS is also used for data quality checks with AMS + BC measurements. The mass closure between both measurements agrees with the correlation (r = 0.93) with a y = 1.11x + 1.06 slope, as shown in Fig. S13.

[Ln 268-270] The coarse mode aerosol influence is also depicted in Figure S4, showing volume distribution from MPSS during the initial heatwave-1 period, which also coincides with high PM mass dominant with larger particles in volume. The GF-PDF illustrates a conspicuous size dependency and temporal variability in the hygroscopic growth of particles.

[Ln 294-296] In some cases, such as on 29th June and 08th July, smaller particles grow as organics mass fraction increases, but no significant change in GF is observed, as shown in PNSD, Figure S4. This is hypothesised to be related to a new particle formation that needs further investigation, as this is not the focus of this study.

[Figure]

Figure S4. The time series of PNSD and volume distribution with Kappa and eBC.

[Figure]

Figure S13. The mass closure for data quality checks with MPSS and AMS + BC mass concentrations.

**Supplementary:** Total mass concentrations (AMS+BC) are calculated, and density correction is applied with the effective density ($\rho$) of individual species (Org, NO3, SO4, NH4, Cl and BC) from literature (Park et al., 2004; Kondo et al., 2011; Poulain et al., 2014). The agreement between mass concentration from both instruments is correlated (r = 0.93) with a slope of y = 1.11x + 1.06, which is suitable for data quality.

**[General Comment]** Page 6, Line 184: Is the 5-10% uncertainty in RH or in GF? Please clarify

**Response:** Thank you for pointing this out. Sorry, the sentence was repeated; it was meant to give uncertainty for kappa's, which is already mentioned in section 2.5. The uncertainty in RH for HTDMA is discussed in section 2.2

We changed and revised the sentence.

[Pg7, Ln 222] Considering the uncertainties in the HTDMA and AMS measurements, the uncertainty between $\kappa_{measured}$ and $\kappa_{chem}$ of 10-15% (leading to an uncertainty of $\pm$ 0.06 for $\kappa$) is roughly estimated, as shown in the supplementary.

**[General Comment]** Section 2.5: The composition of the particles indicates both hydrophilic and hydrophobic compounds present (varied solubility). There are many different variations of $\kappa$ prediction, such as Petters & Kreidenweis 2008 which considers solubility, and Nakao 2017 which uses O:C ratio to consider solubility and $\kappa$. These papers, along with other papers considering low solubility/hydrophobic compounds, found that accounting for the volume fraction dissolved as opposed to ZSR alone can improve $\kappa$ predictions. Is this something that has been considered in $\kappa$chem ?

**Response:** Thank you for highlighting the role of solubility in improving $\kappa$ predictions. The approach we used in $\kappa_{chem}$ primarily follows the Zdanovskii-Stokes-Robinson (ZSR) mixing rule and, indirectly, incorporates solubility and chemical composition information in $\kappa$ calculations. Hence, in ZSR-based $\kappa$ predictions, the values of $\kappa_i$ used in this study (see Table 1) are based on literature values with water activity calculation. Petters et al., 2009; Wex et al., 2009 explain that lower soluble compounds show slight hygroscopic growth and aerosol water contents at relative humidities less than 98%. Changing $\kappa_i$ in calculations doesn't significantly change $\kappa_{chem}$ variations in this study.

We added a sentence and revised the manuscript.

[Pg6, Ln 206 – 220] Throughout subsequent discussions, $\kappa_{measured}$ and $\kappa_{chem}$ denote the Kappa values derived from HTDMA and AMS plus AE33 measurements. The composition in this study indicates the presence of both hygroscopic and hydrophobic compounds, which may affect the particle's overall hygroscopicity. The approach we used to predict hygroscopicity follows the Zdanovskii-Stokes-Robinson (ZSR) mixing rule. To some extent, we indirectly incorporate solubility and chemical composition information in $\kappa$ calculations, particularly for particles with significant fractions of organics more than 50%. Hence, in ZSR-based $\kappa$ predictions, the values of $\kappa_i$ used in this study (see Table 1) are based on literature-measured values, with some water activity calculation. Changing $\kappa_i$ in calculations doesn't significantly change $\kappa_{chem}$ prediction as lower soluble compounds like organics generally show only slight hygroscopic growth but a much better activity as a CCN than indicated by the hygroscopic growth, suggesting highly non-ideal behaviour for aerosol water contents at relative humidities less than 98% (Petters et al., 2009; Wex et al., 2009). The individual values of $\kappa_i$ reflect literature-measured and theoretical predictions as cited in Table 1. i.e., Ammonium nitrate, ammonium sulphate, and organics are based on empirical measurements. Considering the uncertainties in the HTDMA and AMS measurements, the uncertainty between $\kappa_{measured}$ and $\kappa_{chem}$ of 10-15% (leading to an uncertainty of ±0.06 for $\kappa$) is roughly estimated, as shown in the supplementary.

**[General Comment]** Line 189: Is $\kappa_i$ in the ZSR rule for kappa chem determined from measurement or derived from Köhler theory? Please clarify

**Response:** Thank you. We revised and added a sentence to the manuscript. The individual values of $\kappa_i$ used in this study (see Table 1) are primarily based on literature values from prior measurements and studies, some reflecting theoretical predictions. For example, the $\kappa$ values for ammonium nitrate, ammonium sulphate salts, and organics are obtained from empirical studies that integrate measurements and theoretical considerations as cited.

[Pg7, Ln 220] The individual values of $\kappa_i$ reflect literature-measured and theoretical predictions as cited in Table 1. i.e., Ammonium nitrate, ammonium sulphate, and organics are based on empirical measurements.

**[General Comment]** Page 8, Line 245: Please clarify why a GF of 1.2 is used as the cut off for hydrophobic and hydrophilic compounds - theoretically a GF of 1.2 at 90% RH would exhibit a $\kappa$ of ~0.08 (some water uptake). Is there a specific paper that highlights why this is the cut off? If so please cite.

**Response:** We added the description in the manuscript as below. As a GF of 1.2 is not theoretical or empirical, so it depends on the aerosol population for different environments. In our study, the differentiation between two populations (hydrophobic and hygroscopic) was at GF=1.2 from mean GFPDF from HTDMA, which aligns with a few previous studies.

[Pg9, Ln 290- 294] A threshold value of GF 1.2 (red dotted line) is considered a cut-off line or differentiation between hydrophobic and hygroscopic particles for all sizes based on the mean of observed GF-PDF distribution where two modes are distinguished, represents a boundary

where minimal water uptake occurs (Kim et al., 2020; Spitieri et al., 2023). This threshold GF coincides with the typical minimum in the observed GF-PDFs in this study, which also aligns with previous studies (Sjogren et al., 2008; Wu et al., 2013).

**[General Comment]** Page 9, Line 269-270 and Figure 4: Why is a correlation of 0.56 reasonable? This seems a bit low - please clarify.

**Response:** Thank you. I have **already answered #RC1.** We revised and corrected the sentence.

[Pg9, Ln322] The time series of the inorganic volume fraction and GF follow a similar pattern but with differing magnitudes with a correlation (r = 0.56). Here, the outliers are the high hygroscopic events during the campaign period, which may influence the correlation.

**[General Comment]** Page 12, Line 360: Reference that GF-PDF equations are in the supplemental information or include in text directly under methodology as it is a highlight of the results

**Response:** Thanks, it was in the referred paper. Now, we also added the equation in the supplementary and mentioned it in the text. GF-PDF equation for corrected GF by (Sjogren et al., 2008)

$$GF(aw, \kappa) = \left(1 + k \frac{a_\omega}{1 - a\omega}\right)^{1/3}$$

**[General Comment]** Conclusion section: Previous laboratory and modeling studies (e.g., but not limited to Riemer et al., 2019, Razafindrambinina et al., 2022) have highlighted the effects of mixing state on hygroscopicity - how do the conclusions of this work tie into previous hygroscopicity/mixing state studies and add novelty? It would be helpful to strengthen the conclusions with these implications and highlight the uniqueness of this work when it comes to studying mixing state effects on hygroscopicity.

**Response:** Thank you for the comment; we included a sentence and highlighted the uniqueness of this work. This work is focused on the influence of external mixing on hygroscopicity with size-segregated GF. Furthermore, only GF is not enough to consider in the prediction of the hygroscopicity of complex and external mixed particles, but GF-PDF would help to identify and improve the prediction.

[Pg15, 496 - 512] The mean GF-PDF and chemical composition during the ACROSS campaign suggest the air mass was a mixture of complex compounds and externally mixed particles. Occasionally, aerosol source emissions are in close proximity. The external mixing is critical and has a complex composition of atmospheric aerosols. The role of the aerosol mixing state in determining climate-relevant properties such as CCN activity and aerosol optical characteristics (Riemer et al., 2019; Wang et al., 2020) needs advanced measurement techniques and modelling approaches, which may be computationally time-consuming and expensive. Using GF-PDFs to analyse aerosol mixing states adds a new perspective of

understanding how internal and external mixtures impact hygroscopic predictions. This complexity could contribute to the limited agreement in the closure study. Most referred studies use the CCN measurements to derive kappa, but mixing information is lacking in these measurements, whereas external mixing using growth spread can improve the understanding of hygroscopicity. The internally and externally mixed particles exhibit different water-uptake behaviours, and the importance of considering it in predicting aerosol hygroscopicity suggested that using multiple complementary measurement techniques can provide a more comprehensive understanding of aerosol water-uptake properties (Razafindrambinina et al., 2022). Our findings highlight the importance of external mixing, and an approach to growth spread in hygroscopicity can improve the modelling parameters. The inadequacy of the ZSR mixing rule is evident in polluted and mixed aerosol environments, challenging the classical method's applicability for hygroscopicity comparison and prediction. One must consider many assumptions to make it work. It highlights the limitations of the ZSR mixing rule in predicting hygroscopic growth for externally mixed aerosols, which are common in urban to sub-urban areas. It can enhance the characterisation of aerosol hygroscopicity in climate models, as it directly influences the cloud microphysical processes and radiative forcing.

- Conclusion [Pg16, 546-550] These findings emphasise the need for caution when using such chemical-derived hygroscopicity measurements in climate models, especially in regions with fresh aerosol emissions close to the source. The current results of combining hygroscopicity measurements with trajectory cluster analysis and $\kappa_{chem}$ predictions represent a significant value in hygroscopicity prediction. Our findings align with previous studies highlighting the significant influence of the mixing state on aerosol hygroscopicity. Most referred studies use the CCN measurements to derive $\kappa$, but mixing information is lacking in these measurements. In contrast, external mixing using growth spread, especially in a sub-saturated regime, can improve the understanding of particle mixing in hygroscopicity prediction. The particle mixing significantly impacts global climate prediction, mainly based on local and regional models. These improve understanding of aerosol growth in sub-saturated conditions and their broader impact on atmospheric processes, including aerosol microstructure and optical properties.